# Imaging of Acute Abdominopelvic Pain in Pregnancy and Puerperium—Part II: Non-Obstetric Complications

**DOI:** 10.3390/diagnostics13182909

**Published:** 2023-09-11

**Authors:** Gabriele Masselli, Giacomo Bonito, Silvia Gigli, Paolo Ricci

**Affiliations:** 1Department of Emergency Radiology-Policlinico Umberto I Hospital, Sapienza University of Rome, Viale del Policlinico 155, 00161 Rome, Italy; gabriele.masselli@uniroma1.it (G.M.); paolo.ricci@uniroma1.it (P.R.); 2Department of Diagnostic Imaging, Sandro Pertini Hospital, Via dei Monti Tiburtini 385, 00157 Rome, Italy; silvia.gigli@aslroma2.it; 3Department of Radiological, Oncological and Pathological Sciences, Policlinico Umberto I, Sapienza University of Rome, Viale Regina Elena 324, 00161 Rome, Italy

**Keywords:** acute abdominopelvic pain, pregnancy, post partum, ultrasonography, computed tomography, magnetic resonance imaging, non-obstetric complications

## Abstract

Emergency imaging in pregnancy and puerperium poses unique challenges both for clinicians and radiologists, requiring timely and accurate diagnosis. Delay in treatment may result in poor outcomes for both the patient and the foetus. Pregnant and puerperal patients may present in the emergency setting with acute abdominopelvic pain for various complications that can be broadly classified into obstetric and non-obstetric related diseases. Ultrasonography (US) is the primary diagnostic imaging test; however, it may be limited due to the patient’s body habitus and the overlapping of bowel loops. Computed tomography (CT) carries exposure to ionising radiation to the foetus, but may be necessary in selected cases. Magnetic resonance imaging (MRI) is a valuable complement to US in the determination of the etiology of acute abdominal pain and can be used in most settings, allowing for the identification of a broad spectrum of pathologies with a limited protocol of sequences. In this second section, we review the common non-obstetric causes for acute abdominopelvic pain in pregnancy and post partum, offering a practical approach for diagnosis and pointing out the role of imaging methods (US, MRI, CT) with the respective imaging findings.

## 1. Introduction

Acute abdominal and pelvic pain during and just after pregnancy poses unique diagnostic and therapeutic challenges, owing to a variety of multiple confounding factors related to gestation and puerperium.

Physiologic leukocytosis in pregnant patients, for instance, and a significant overlap between symptoms that occur as part of a normal pregnancy and those due to underlying disease, make clinical assessment difficult [1].

The enlarging uterus displaces the adjacent organs from their normal locations, limiting physical examination; therefore, the classic clinical semeiotic in acute pain is altered [2].

A wide range of conditions that are associated with or unrelated to pregnancy may cause abdominopelvic pain, making differential diagnosis extensive. The delay in diagnosis and intervention may lead to poor outcomes for both the mother and the foetus.

The puerperium period, which may take as long as 8 weeks from birth, is also burdened by a wide clinical spectrum of complications, significantly associated with the delivery method. A multimodality imaging approach is often required to clarify an indeterminate clinical scenario in pregnancy and puerperium, playing an important role in the assessment of complications and expediting diagnosis.

Ultrasonography (US) is the first line of imaging investigation both during and following pregnancy, because of its availability, portability and lack of ionizing radiations. However many causes of acute abdominal pain are not readily diagnosed with US, especially in later stages of gestation [3]. The drawbacks of US are overcome by magnetic resonance imaging (MRI), a powerful radiation-free diagnostic tool, the use of which is growing in the emergency setting due to its increased diagnostic accuracy [4]. Diagnostic modalities that employ ionizing radiation, such as computed tomography (CT), should be avoided in pregnancy for fetal safety issues; however, when deemed necessary, the use of CT should not be delayed because of the fear of ionizing radiation [5,6]. Although of limited use in pregnancy, CT is the imaging modality of choice to investigate post-partum acute complications in doubtful cases and/or when US is inconclusive [7]. This article aims at describing the imaging findings of common and uncommon causes of acute abdomino-pelvic pain, occurring during and just after pregnancy. The previous instalment reviewed the main obstetric (non-fetal) causes of abdominopelvic pain, including ectopic pregnancy, placental abruption, placenta percreta, uterine rupture, post-partum haemorrhage, retained products of conception and endometritis. The present second section addresses the main non-obstetric emergencies associated with acute abdominopelvic pain in pregnant and puerperal patients, including the gastro-intestinal, hepatic, pancreatobiliary, urogenital and vascular etiologies summarized in Appendix A.

## 2. Gastrointestinal Diseases

### 2.1. Acute Appendicitis

Acute appendicitis (AA) is the leading cause of acute abdominopelvic pain during pregnancy, occurring in about 0.07% of cases (1 in 1500 pregnancies), most often in the second trimester. It also represents the most common non-obstetric surgical emergency in pregnancy [2]. Diagnosis may be challenging and delayed, as clinical presentation is confounded by several factors including physiologic leukocytosis, the displacement of the inflamed appendix by the growing uterus away from the right iliac fossa, a difficult abdominal examination, and non-specific symptoms; up to 25% of pregnant women with AA do not experience right iliac fossa pain [2]. Timely and accurate diagnosis is crucial because of the risk of fetal loss, which is around 6% if the maternal appendix perforates, while it is around 1.5% for uncomplicated forms [8]; a surgical delay of longer than 24 h results in a 66% increase in the rate of appendiceal perforation [9]. Therefore, imaging is aimed at reducing delays in diagnosis and treatment.

According to ACR Appropriateness Criteria, US with graded compression is the first imaging modality for pregnant patients with suspected AA, as it does not involve ionizing radiation [10]. The diagnostic accuracy of US for detecting AA in pregnancy widely varies in the literature, with reported values of sensitivity and specificity ranging from 50 to 100% and from 33 to 92%, respectively [6,11].

A recent meta-analysis, including 1593 patients, showed a relatively low diagnostic accuracy of US for AA during pregnancy, with a sensitivity of 77% and a specificity of 75% [12]. The sensitivity of US in the first, second and third trimesters was 69%, 63% and 51%, respectively, while the corresponding values for specificity were 85%, 85% and 65% [12]; there was a significant decrease in the diagnostic performance of US between the first two trimesters and the third. Moreover, a retrospective study by Lehnert et al. showed that US did not detect AA in 71% of cases during the second and third trimesters with surgically proven disease, emphasizing that appendix evaluation is limited by the patient body habitus, especially in the later stages of gestation, as well as by the clinician’s expertise [13].

As in non-pregnant patients, AA sonographically appears as a blind-ending, dilated (>6–7 mm in diameter), peristaltic, non-compressible and thick-walled (>2 mm) tubular structure arising from the cecum. Indirect features include fluid effusion in the right iliac fossa and periappendiceal fat stranding (Figure 1).

Segev et al. showed that if the appendix is detectable and diagnostic criteria for AA are met, the positive predictive value of US is 94%. However, the negative predictive value is low (40%) [14]. Therefore, a negative US should not be taken as a definitive finding, and MRI should be considered in a patient with clinical suspicion [10].

Magnetic resonance imaging is a valuable tool for diagnosing AA in pregnant women due to its lack of ionizing radiation when compared to CT with high sensitivity and specificity of 91.8% and 97.8%, respectively [2]. Fonseca et al. reported a retrospective review of the records of 79 pregnant patients who underwent sonography, while 31 of those also had an MRI. All pathways resulted in high specificity, whereas sensitivity was 25% for clinical diagnosis, 39% for US and 100% for MRI [15]. The authors also demonstrated that the patients undergoing MRI were more frequently discharged from the emergency department and had a shorter length of stay than patients without MRI [15]. A meta-analysis evaluating the diagnostic performance of MRI in the setting of AA during pregnancy reported a sensitivity and specificity value of 96.8% and 99.2%, respectively [16].

One of the major advantages of MRI is its 100% negative predictive value for the diagnosis of AA in pregnancy [17]. Rapp et al. observed that the routine integration of MRI into the clinical work-up for suspicion of AA, during pregnancy, is significantly associated with a lower negative laparotomy rate (from 55% to 21%), without a change in the perforation rate [18]. Moreover, MRI can identify conditions mimicking AA in pregnant patients, such as obstructing urinary tract calculi, pyelonephritis, ovarian torsion and other GI conditions. Therefore, some authors have proposed MRI as the first-line imaging modality for the evaluation of AA during pregnancy [17,19].

The MRI findings for AA mirror those of other modalities, including an enlarged appendix (diameter greater than 7 mm), with a thickened wall due to mural oedema (>2 mm), high signal intensity luminal contents on T2 weighted-images (T2WI) and periappendiceal fat stranding or free fluid [20]; the appendicolith may appear as a round-shaped structure hypointense on all pulse sequences (Figure 2).

In DWI, the inflamed wall may exhibit restricted diffusion both in perforated and non-perforated cases [21]. MRI may also depict complications such as perforation and periappendiceal abscess. If MRI cannot be performed, due to absolute contraindications or unavailability, the supplementary use of CT may avoid both unnecessary surgery as well as delays in diagnosis and treatment. Positive CT features of AA are the same as in the non-pregnant population, with high sensitivity and a specificity of 92% and 99%, respectively [11].

Surgical intervention is recommended in the first 24 h in pregnant patients after the diagnosis of AA to avoid complications, such as perforation.

### 2.2. Small Bowel Obstruction

SBO is a common clinical entity that is extremely rare in pregnancy, with an incidence rate of 0.001–0.003%, carrying a significant risk to mother and foetus: Webster et al. reported a maternal mortality rate of 2% and a fetal loss of 17% [22]. SBO in pregnancy is most commonly attributed to adhesions from previous abdominal surgery (around 60% of cases), followed by internal hernia (15%), intussusception (12%) and volvulus (9%). Young women of childbearing age undergoing bariatric surgical procedures or restorative proctocolectomy for inflammatory bowel diseases (IBD) or familiar cancer syndromes are at increased risk for adhesive disease or internal hernia (IH)) [22,23,24,25]. The US is the first mode in the evaluation of bowel disease in pregnancy, showing dilated and fluid-filled loops with levels and aperistalsis; however, the detection of the transition point and the causes of bowel obstruction often remains undetermined. MRI, performed with the use of multiplanar T2-weighted single-shot fast spin-echo (SSFSE) imaging, is useful in both detecting and characterizing SBO in pregnancy, because of its lack of ionizing radiation and excellent soft tissue resolution. Small bowel loops should be traced to the transition point between the dilated proximal and decompressed distal bowel. This transition point should be assessed to detect the cause of the obstruction [3,20,26,27] (Figure 3).

The diagnosis of IH is clinically challenging, and the challenge is exacerbated in pregnancy. The literature on the imaging of IH, specifically in pregnant women, is scarce. Krishna et al. evaluated the utility of MRI in the diagnosis of IH in pregnant women who had previously undergone Roux-en-Y Gastric Bypass (RYGB); the authors showed that the signs with the best interobserver agreement and diagnostic odds ratio were mesenteric swirl, SMV beaking, mesenteric vascular congestion and mesenteric edema.

This study reported that MRI has a comparable specificity to CT (86–100%) and a lower sensitivity (75–88%), concluding that MRI might be a reasonable and safe alternative to CT [28]. In contrast, Bonouvrie et al. reported a lower diagnostic accuracy of MRI (for SBO in pregnant women after bariatric surgery) with a sensitivity and specificity of 67% and an NPV of only 22%, and significant variability in the interobserver agreement. Furthermore, the authors observed that MRI is unable to detect SBO in almost one in three patients (29%), concluding that this method should only be performed in cases of mild clinical presentation [29]. Otherwise, in a severe scenario, diagnostic laparoscopy remains the gold standard [22,29].

### 2.3. Inflammatory Bowel Disease

IBDs, including Crohn’s disease and ulcerative colitis, commonly affect women in the reproductive-aged population, with a peak incidence between 15 and 25 years. Most pregnancies in patients with quiescent IBD are uncomplicated. When IBD is exacerbated, it usually presents with fever, bloody diarrhoea and abdominal pain [30]. IBD is most likely to affect the terminal ileum (80% of cases), mimicking appendicitis.

MRI is the preferred modality to assess active IBD and its complications in pregnancy, with a sensitivity and specificity ranging from 88% to 98% and 78% to 100% [3]. IBD-related imaging features of active disease mirror those in non-pregnant women. Mural findings include the bowel wall circumferentially thickening to greater than 3 mm, mucosal ulcerations, submucosal oedema (resulting in signal hyperintensity on T2-weighted images) and bowel lumen narrowing with upstream dilatation. Mesenteric findings include engorgement of the vasa recta (comb sign), fibrofatty proliferation (creeping fat) and reactive lymphadenopathy. A disadvantage of MRI lies in its lower sensitivity to extraluminal air compared to CT; the identification of extraluminal susceptibility artifacts that bloom on in-phase sequences could be useful. MRI is also helpful in evaluating complications, including abscesses, fistulae and chronic bowel strictures [3,20,26].

## 3. Hepatic and Pancreatobiliary Diseases

### 3.1. Pregnancy-Related Liver Diseases

Pregnancy-related liver diseases (PLDs) represent the most frequent cause of liver dysfunction in pregnancy and affect up to 3% of pregnant women. When severe, these conditions are associated with significant morbidity and mortality for both the mother and the foetus [31].

PLDs exhibit trimester-specific characteristics, whereas non-pregnancy-related liver diseases can occur at any time [32]. The timing of the occurrence is crucial for diagnosis and treatment strategies.

PLDs include intrahepatic cholestasis of pregnancy (ICP), acute fatty liver of pregnancy (AFLP) and hemolysis, elevated liver enzymes and low platelets count (HELLP) syndrome. In addition, pre-eclampsia (PE) and hyperemesis gravidarum (HG) are frequently associated with liver abnormalities [32,33].

Transabdominal US and MRI can be safely performed in pregnant women to evaluate the liver. In doubtful cases, the use of computed tomography (CT) could be required, involving radiation exposure to the foetus.

#### 3.1.1. Intrahepatic Cholestasis of Pregnancy

ICP, also known as obstetric cholestasis, is the most common pregnancy-specific liver disorder, with a reported incidence ranging from 0.2% to 2% [34]. ICP is characterized by cholestasis and pruritus, with typical onset in the late second or third trimester of pregnancy. It is associated with abnormal liver function in the absence of other liver diseases, and with the spontaneous resolution of symptoms and biochemical abnormalities after delivery. It has been supposed that the role of estrogen and progesterone in the development of cholestasis is related to a decrease in hepatic biliary transport protein expression and an internalization of the bile salt export pump [35]. At present, the most sensitive biochemical marker in the diagnosis of ICP is the level of total bile acids, with a cut-off value of 10 µM/L [36]. The risk of fetal complications, including stillbirth, respiratory distress syndrome, meconium passage and fetal asphyxiation, increases in severe cholestasis when the serum bile acid level exceeds 40 µM/L [36].

US is the method of choice; the liver usually shows a normal echotexture, but in the presence of biliary symptoms, US may be useful to rule out other causes of obstructive biliary tract pathology [6].

#### 3.1.2. Acute Fatty Liver of Pregnancy

Acute fatty liver of pregnancy (AFLP) is a rare and potentially fatal disorder for both the mother and the conceptus, resulting from the micro-vesicular fatty infiltration of hepatocytes, which can lead to liver failure [37].

It usually occurs in the third trimester with an incidence of 1/7000 to 1/16,000 obstetric emergencies, or in the early post-partum period [37]. The most common symptoms are nausea, vomiting, polyuria, polydipsia, abdominal pain, jaundice and encephalopathy.

Currently, the diagnosis and management of AFLP is based on clinical and laboratory findings alone in almost all patients. Hypoglycaemia is a poor prognostic sign. Imaging could show signs of fatty infiltration of the liver, although current diagnostic modalities have limited utility in the setting of AFLP. US of the liver may demonstrate a hyperechoic appearance of the hepatic parenchyma from fatty infiltration. MRI performed with T1 dual gradient-echo in-phase (IP) and out-of-phase (OOP) sequences can show a drop in liver signal intensity on the OOP images, suggestive of steatosis [6].

#### 3.1.3. HELLP (H = Haemolysis, EL = Elevated Liver Enzymes, LP = Low Platelets) Syndrome

HELLP syndrome is a variant of severe pre-eclampsia that occurs in 0.2–0.6% of all pregnancies and in up to 12% of patients with pre-eclampsia [38]. This condition is characterized by the association of three laboratory features: hemolysis (H), increased liver enzyme levels (EL) and low platelet count (LP). Approximately 70% of cases are diagnosed in the antepartum period (between 27–36 weeks of gestation), whereas about 30% of cases occur after delivery, often within the first 48 h [39]. Risk factors for HELLP are advanced maternal age and multiparity.

It can be classified into mild, moderate or severe depending on the alanine aminotransferase (ALT), lactate dehydrogenase (LDH) levels and platelet count, and as early (<34 weeks) or late (≥34 weeks) according to the gestational age at diagnosis or delivery [37].

There is no specific symptom or sign that differentiates HEELP syndrome from pre-eclampsia. Right upper quadrant abdominal pain is the most common feature. Other vague symptoms include nausea, vomiting and headache. Diagnosis is mainly based on clinical presentation and laboratory findings [37,39]. In a retrospective review including 568 patients with pre-eclampsia or HELLP syndrome, only 0.53% of them had abnormal hepatic imaging features [40].

However, diagnostic imaging should be recommended in patients with suspected HEELP syndrome, and who also report abdominal pain, to rule out life-threatening complications such as hepatic parenchymal hematoma or infarction, and hepatic rupture secondary to necrosis or haemorrhage [38,41]. US and MRI should be preferred in pregnancy due to the absence of ionizing radiation, while CT is the method of choice in the post-partum period, especially in unstable patients. Even though US can be quickly carried out, CT or MRI may be better in detecting hepatic abnormalities, such as hematomas, or in assessing the extent of the intraperitoneal haemorrhage in cases of hepatic rupture [6].

Sonographic findings are often nonspecific, including enlarged liver (predominantly the right lobe), peri-portal edema and free abdominal fluid. On US, hepatic hematoma can appear as a peripheral area, heterogeneously hypoechoic as well as isoechoic, relative to the uninvolved liver; the diagnosis has to be further confirmed by cross-sectional imaging [41]. CT and MRI appearance of hepatic hematoma vary according to the age of the bleeding, but are usually depicted as heterogeneous space-occupying lesions that compress the adjacent parenchyma. Recent haematoma typically appears as hyperattenuating relative to neighboring parenchyma [38]. CT can accurately determine the size of the hematoma and can detect the active extravasation of contrast medium, suggestive of active bleeding [42]. Nunes et al. observed that hematomas most frequently involved the right lobe and the medial section of the left lobe [40].

Liver infarction can be undetectable through US or appear as peripheral geographic hypoechoic bands [43]. MRI can show corresponding parenchymal signal changes, such as hyperintense on T2WI and hypointense on T1WI, according to the degree of the necrosis [44]. CT shows peripheral, ill-defined and wedge-shaped areas of decreased or absent enhancement, without mass effect (Figure 4); enhanced intrahepatic vessels are visible within the ischemic areas [42,45].

The development of hepatic capsular rupture is a rare but catastrophic event, occurring in less than 2% of patients with HELLP syndrome [39]. Patients may present with acute abdominal pain and haemorrhagic shock. US demonstrates complex free fluid within the abdominal cavity, suggestive of haemoperitoneum. A hepatic fragment floating in the peritoneal effusion has been reported [46]. CT typically depicts hepatic rupture as a site of focal capsular irregularity with associated liver hematoma and/or hemoperitoneum; in patients with active bleeding, CT reveals the spotty extravasation of iodinated contrast agent [38]. Hepatic arteriography can establish the site of hemorrhage and is only performed before arterial embolization [30].

### 3.2. Acute Cholecystitis and Cholelithiasis

Acute cholecystitis (AC) is the second most common non-obstetric indication for surgery during pregnancy, often resulting from an impacted stone in the cystic duct. The prevalence of gallstones during pregnancy is higher than 12%, whereas symptomatic biliary disease is uncommon, occurring in 0.1–0.3% of patients [47].

Pregnancy is a known risk factor for the development of gallbladder disease due to the increased levels of circulating estrogen and progesterone, which lead to cholestasis and supersaturation of bile with cholesterol [47]. AC is the most common complication of cholelithiasis and has been reported to occur in the setting of intrahepatic cholestasis of pregnancy [31]. Other complications of biliary tract obstruction mimic those in the non-pregnant population, including choledocholithiasis, pericholecystic abscess, gangrenous and emphysematous cholecystitis, gallbladder perforation and acute pancreatitis [48].

As in the general population, a right upper quadrant US is the first imaging modality of choice when the clinical presentation is suggestive of biliary pathology [49]. US typical findings include gallbladder distension (short-axis diameter > 3 cm), mural thickening (>3 mm), pericholecystic fluid, gallstones, if present, and a positive sonographic Murphy’s sign (maximal abdominal tenderness when the US probe is applied over the gallbladder) [50]. The latter has been attributed to high specificity (88.3%) and sensitivity (71.9%) in isolation and even higher values when associated with cholelithiasis [51]. Childs et al. confirmed the high specificity (98%) of sonographic Murphy’s sign, but sensitivity was extremely poor (19%) [50]; therefore, the absence of this sign should not prevent the radiologist from making a diagnosis of AC if the other aforementioned US findings are detected.

In a meta-analysis of 5859 patients with AC in the general population, the sensitivity and specificity of US has been reported at 81% and 80%, respectively; the author concluded that the diagnostic accuracy of US has a substantial margin of error, comparable to that of MRI [52].

Although US usually provides the definitive diagnosis of AC, MRI should be employed as a second-line tool in equivocal cases for further assessment of the biliary tree and gallbladder, or to confirm clinal suspicion of complications [49]. MR cholangiopancreatography (MRCP) has been shown to be more accurate than non-contrast CT in diagnosing AC, with a positive predictive value of up to 100% [53], and in detecting biliary stone disease, with high sensitivity and specificity (98% and 84%, respectively) [54].

MRI findings of AC include hydropic distention, gallbladder wall thickening with edematous stratification and pericholecystic inflammatory changes; these features are best imaged on fluid-sensitive T2-weighted fat-suppressed sequences (Figure 5).

MRCP offers an excellent depiction of the biliary system, gallbladder and pancreatic duct. Stones are shown as rounded or angular geometric filling defects, easily detectable on thin-cut sequences, against the hyperintense signal of the bile [55]. MRI enables differential diagnosis between choledocholithiasis and intrahepatic cholestasis of pregnancy, characterized by an overlapping clinical presentation [44].

Moreover, MRI allows for the identification of any complications of AC. In gangrenous cholecystitis, MRI shows asymmetric thickening and irregularity of the gallbladder wall with increased signal intensity on T2WI; other signs of gangrenous cholecystitis, including ulceration, haemorrhage, necrosis or microabscesses in the gallbladder wall, are best depicted through MRI [56]. The finding of gas within the gallbladder wall or lumen, which appears as signal voids, is suggestive of emphysematous cholecystitis. Gallstones can be distinguished from intraluminal gas by their location; gas is shown as multiple signal voids floating in the non-dependent portion of the gallbladder and or within the biliary tract, whereas stones are typically identified in the dependent portion [57].

Clinical practice guidelines by the Society of American Gastrointestinal and Endoscopic Surgeons (SAGES) recommend early surgical management with laparoscopic cholecystectomy for pregnant women with symptomatic gallbladder disease, regardless of trimester [58].

### 3.3. Acute Pancreatitis

Acute pancreatitis in pregnancy is a rare, usually self-limiting condition, estimated to occur in every 1/1000 to 1/5000 cases, often in the third trimester [59]. Maternal and fetal mortality rates have decreased over the years, from 37% to 3.3% and from 60% to 11.6%, respectively, due to progress in diagnosis and treatment [60]. Acute pancreatitis in pregnancy is most commonly caused by cholelithiasis; several factors contribute to gallstone and sludge formation in pregnant women, such as increased cholesterol synthesis, bile stasis and decreased gallbladder contraction [47]. The symptoms are similar to those in nonpregnant patients, with acute onset epigastric pain classically radiating to the back, nausea, vomiting and low-grade fever. Laboratory abnormalities suggesting acute pancreatitis are elevated serum amylase and lipase, which are usually increased to at least three times the upper limits of normal (normal values: lipase, 30–210 IU/L; amylase, 30–110 IU/L) [61].

US of the right upper quadrant is aimed to confirm or rule out biliary etiology; it is a safe, reliable and cost-effective method but its use is limited by the patient’s body habitus, especially during the third trimester of gestation, and/or by the overlap of bowel gas [3]. If US is normal or indeterminate, MR imaging combined with MRCP should be performed [62]. This technique enables the assessment of the pancreatic parenchyma, peripancreatic tissues and the biliary tract, with a sensitivity greater than 90% [54,63,64].

MRI demonstrates pancreatic enlargement and oedema, with reduced intensity on T1-weighted images and increased intensity on T2-weighted images (Figure 6).

On DWI, acute early pancreatitis shows restricted diffusion and lower ADC values than the spared parenchyma [65]. In severe pancreatitis, necrotic foci within the parenchyma are hypointense on T1-weighted images and hyperintense on T2-weighted images, compared to the non-necrotic gland. Parenchymal haemorrhage can be detected at spotted or patchy high-signal intensity (like “salt”) on T1-weighted fat saturated sequences. Peripancreatic abnormalities, such as fluid collection, edema or fat stranding are better identified on fluid-sensitive sequences, appearing as high-intensity signals surrounding the gland. MRCP sequences acquired with respiratory gated-thin sections, are best suited to depict biliary and pancreatic ductal dilatation, allowing the detection of gallstones, shown as filling defects in the gallbladder and biliary tract [3,41,66]. The majority of pregnant patients with AP do not have complications. Gilbert et al. reported a complication rate of 1.44%, all of which were acute peripancreatic fluid collections (APFC) [67]. APFC occur in 50% of interstitial edematous pancreatitis. They are usually rounded with thin walls, often located in the lesser sac or anterior pararenal space. MRI is better than CT for detecting solid content or internal hemorrhage in APFC [68]. Vascular complications include arterial pseudoaneurism and venous thrombosis, most commonly involving splenic vessels; these abnormalities are typically assessed on contrast-enhanced MR angiography in non-pregnant populations, but can also be assessed with flow sensitive time-of-flight angiographic techniques [66].

## 4. Urogenital Tract Diseases

### 4.1. Urolithiasis and Pyelonephritis

Physiologic hydronephrosis is the most significant change in the urinary tract during pregnancy, occurring in up to 90% of patients, especially in primigravidas. It is an asymptomatic condition, secondary to increased progesterone, which causes ureteral smooth muscle relaxation, and to the compression of the ureter between the gravid uterus and the ileopsoas muscle [69]. Physiologic hydronephrosis is most commonly right-sided and is observed almost exclusively in the third trimester, with spontaneous resolution several weeks post partum [70].

Renal colic represents severe pain associated with obstructive hydronephrosis, most often caused by a kidney stone; it is the leading non-obstetric cause of abdominal pain and subsequent hospital admission during pregnancy [71]. The incidence of a symptomatic stone event is relatively rare, occurring in about 1 in 2000 pregnancies, and similar to the childbearing non-pregnant population [72,73].

Maternal physiologic changes, such as the aforementioned physiologic hydronephrosis, make it challenging for the urologist to rely on traditional clinical signs and symptoms of urolithiasis for diagnosis, complicated by the concern of avoiding exposing the conceptus to radiation from CT.

Ureteral obstruction may lead to significant maternal morbidity, including pyonephrosis, urosepsis, preterm labor, cesarean delivery, hypertensive disorders and gestational diabetes; these potential complications make timely and accurate diagnosis crucial so that proper management can be initiated for the health of the pregnant patient and the foetus [71].

Ultrasonography (US) represents the primary radiological investigation of choice to evaluate for suspected urolithiasis in pregnancy, due to its lack of ionizing radiation, low cost and availability. However, several limitations, such as the patient’s body habitus, the overlap of bowel gas and dependence on the operator, reduce the diagnostic accuracy of this method; the performance in terms of sensitivity for urolithiasis is quite low (11–24%), especially in cases of middle ureteric stone [74]. Moreover, in the absence of a definitively visualized stone, differentiating between ureteric obstruction and physiological pregnancy-related hydronephrosis can be limited, with a consequent increase in false positives. However, the finding of a dilated infra-iliac ureter should suggest the presence of a low ureteral stone, excluding gestational hydronephrosis [75]. Transvaginal US has been shown to be useful in evaluating the distal ureter, as well as the differentiation between obstruction and physiologic hydronephrosis, when abdominal US is inconclusive; however, the lack of availability and expertise may limit its employment [76].

Doppler ultrasound findings, such as differences in the resistive index (RI), the absence of ureteral jets and twinkling artefact, can help to increase diagnostic accuracy in detecting ureteral stones. RI does not appear to be affected by gestational hydronephrosis, providing an indirect estimate of the renal perfusion that can be reduced in the setting of obstructive uropathy, due to back pressure on the collecting system; an elevated intrarenal RI (≥0.70) and a mean delta RI value (ΔRI: interrenal difference in RI) falling in the range of 0.04–0.08, between the symptomatic kidney and the contralateral one, raise suspicion of an obstructive process [77].

The absence of a “ureteral jet” (passage of urine at the ureterovesical junction) on the symptomatic side may increase sensitivity and specificity to 100% and 91%, respectively. However, patients should be scanned in the contralateral decubitus position to decrease false-positive results [78].

Magnetic resonance urography (MRU) is a non-invasive and non-operator-dependent method, representing the second-line test for suspected urolithiasis in pregnancy, when US is equivocal [75].

The use of heavily T2-weighted or water-weighted images with thick slabs is helpful in depicting the urinary system, enabling differential diagnosis between physiological and obstructive hydronephrosis [6,79].

MRU findings in physiological dilatation include a lack of visible filling defects and a collapsed ureter below the pelvic brim; there is a characteristic smooth tapering of the middle third of the ureter, due to the mass effect between the gravid uterus and adjacent retroperitoneal musculature [6,74].

Urolithiasis produces a different MR pattern; an acutely obstructed kidney is asymmetrically enlarged and edematous, showing an increased T2-weighted signal intensity of the parenchyma. The MR urography appearance of a “double kink sign”, with constriction at the pelvic brim and the vescicoureteral junction with a standing column of urine in the pelvic ureter, suggests an obstructing distal ureteral stone [75].

Other features that are indicative of pathological hydronephrosis include an “unusual” site of obstruction, an abrupt ending of the ureter (rather than smooth tapering at the level of the pelvic brim) and perirenal or periureteral fluid (reflecting lymphatic distension and fluid leakage from the obstructed kidney) [6,75].

A filling defect within the lower ureter is a direct sign of obstruction, although the sensitivity of MRI for stone detection may be limited by calculus size; the use of thin slice (3 mm), high-resolution T2 fast spin echo (FSE) sequences improves the accuracy in identifying small stones [3,6,79].

The impaired egress of urine (urinary stasis), due to stones, increases the risk of complications, such as acute pyelonephritis, reported in 1% of pregnancies [80]; in these cases, MRI could be a useful adjunct to clinical diagnosis. The DWI sequence in particular has been found to have increased sensitivity when compared to non-contrast CT (95% and 67%, respectively). The kidney appears enlarged and edematous with areas of lower signal intensity on T2-weighted images and restricted diffusion on DWI (Figure 7).

In patients who develop renal abscesses, T2-weighted and DW images respectively show a focal, more hyperintense signal and restricted diffusion, compared to the rest of the parenchyma involved in nephritis [3,54] (Figure 8).

MRU has several drawbacks (when compared with non-contrast CT) because of its lower spatial resolution, prolonged imaging times, limited availability, poor sensitivity to detect small stones and interference with metal objects [81]. Further limitations include spatial misregistration, which may occur in the patient freely breathing and the presence of flow-void artefacts in urine. These artefacts, which may mimic filling defects, are typically nondependent [3,74,75].

The usefulness and benefit of low-dose CT for evaluating suspected urolithiasis has been confirmed in the non-gravid population; CT showed superior sensitivity and specificity (98% for both) to other diagnostic techniques in localizing urinary tract stones, providing information for causes of non-urologic flank pain. However, diagnostic modalities that use ionizing radiation should be avoided in pregnant patients, owing to risks to the foetus. Low-dose CT should, therefore, be considered in unresolved cases as a last-line test alternative to MRI, only during the second and third trimesters, when the foetus is relatively less radiosensitive compared to the first trimester [82,83].

Overall, recent instances in the literature reported that the positive predictive value of CT, MRI and US is 95.8%, 80% and 77%, respectively, for the detection of urolithiasis during pregnancy [81,84].

### 4.2. Ovarian Torsion and Adnexal Masses

Ovarian torsion (OT) is the fifth most common gynecological surgical emergency, with a reported incidence of 2–3%. However, during pregnancy it is a rare event, occurring in 1 in 800 patients, more commonly in the late first to early second trimesters, when the uterus is rapidly enlarging [85,86]. Some studies have reported that assisted reproductive technologies are a major risk factor for OT in pregnancy [87].

The torsion may occur in the normal ovary, when it twists on its ligamentous supports, but it is usually secondary to a preexisting adnexal mass, including hemorrhagic cysts, simple cysts, endometriomas and benign tumours (i.e., teratomas, cystadenomas and adenofibromas). Clinical features are non-specific, with intermittent abdominal pain, nausea and vomiting; most patients are symptomatic (94–100%). Early diagnosis and surgery are necessary to avoid adnexal necrosis, and preserve ovarian function in the patient [88].

US combined with color Doppler imaging is usually the first imaging tool for the diagnosis of OT: it should be suspected in any pregnant patient with an adnexal mass and lower abdominal pain. A unilateral enlarged ovary (>5 cm) with or without a pre-existing mass, multiple peripherally displaced follicles with echogenic stroma, vascular pedicle twisting (whirlpool sign) and free pelvic fluid, is reported to be a common ultrasonographic finding in OT. Ovarian edema is most common among pregnant patients with OT compared to non-pregnant women [89]. Doppler flow imaging could improve the diagnostic accuracy of OT in non-pregnant patients, showing the absence/decreased blood flow; however, during pregnancy, this method shows decrease sensitivity and higher false negative rates [90]. Therefore, the persistence of ovarian blood flow does not rule out torsion; this could be explained by the progressive nature of OT. When the torsion is of low grade, it will first impair the venous and lymphatic drainage, making the arterial waveforms detectable [91]. After inconclusive US, MRI represents a better technique by which to assess suspected OT, depicting both direct and indirect findings more clearly [3,92]. A retrospective study, evaluating the diagnostic accuracy of MRI for OT in pregnancy, reported sensitivity, specificity, positive and negative predictive values of 100%, 77.8%, 90.5% and 100% [93].

The pathognomonic and most specific feature is a twisting of the ovarian pedicle, presenting as a beak-like protrusion adjacent to the ovarian mass or to the enlarged ovary. However, it is only identified in less than one-third of patients through CT and MRI [92,94]. In the early stages, an asymmetric enlarged oedematous ovary with central follicular stromal and peripherally displaced follicles in a string-like appearance (“pearl string sign”) is well demonstrated by MRI, especially on T2-weighted sequences. In the later stages, haemorrhagic alterations develop, associated with infarction and secondary necrosis of the adnex; subacute haemorrhage can be detected as hyperintense on fat-saturated T1-weighted images, involving the periphery of the ovary or medullary stroma.

In this stage, gadolinium administration shows heterogeneous, minimal or absent perfusion, confirming the evolution toward necrosis; however, the administration of contrast medium should be avoided in pregnancy. Other findings include ipsilateral uterine deviation, a blood-filled fallopian tube (haematosalpinx) and a haemoperitoneum [92,93,94,95]. DWI with both visual assessment and quantitative ADC measurement may be helpful in the diagnosis of OT, especially in pregnant women. Bekci et al. showed that the mean ADC value of the torsion ovary was significantly lower than that of the nonaffected side [96]. Kato et al. observed that ADC values were significantly lower in patients with haemorragic infarction than in those without [97].

MRI also represents the “gold standard” in the assessment of adnexal masses that remain indeterminate through US, having the best potential for the preoperative evaluation of these lesions. It has shown greater accuracy (88.9%) than US (63.9%) in the characterization of adnexal masses as malignant and a higher specificity (83.7% vs. 39.5%) [3,98,99,100].

MRI provides specific information about tissue content, including the presence of fluid, blood, fat and collagen. Signal hyperintensity on T1- and T2-weighted sequences is suggestive of intralesional fat or subacute blood products; in these cases, T1-weighted frequency-selective fat saturation pulse sequences should be obtained, enabling differential diagnosis between mature fat-containing teratomas and hemorrhagic cysts or endometriomas. Fluid-filled lesions show signal hypointensity on T1-weighted images and hyperintensity on T2-weighted images, as seen in functional cysts and cystic tumours. Masses showing low signal intensity on T1- and T2-weighted images likely contain collagen or hyalinized tissue, such as uterine leiomyoma, ovarian fibrotecoma and Brenner tumours [6,92,101,102].

Features such as prominent solid components within a cystic mass, necrosis in a solid mass or peritoneal implants are consistent with ovarian malignancies. Any cystic lesion should be carefully evaluated for mural nodules or septal thickening, as well as papillary projections, because these findings are suggestive of malignancy [101,102,103].

Most ovarian masses diagnosed in pregnancy are benign and are spontaneously resolved (Figure 9). Surgical management is warranted when masses are suspicious for malignancy, at risk for torsion or clinically symptomatic [3].

### 4.3. Uterine Leiomyoma

Uterine leiomyoma is the most common tumour of the female reproductive tract, affecting 70–80% of patients during their lifetime. The prevalence of leiomyomas in pregnancy is estimated to exceed 11%, with higher frequency in advancing maternal age [104,105].

Leiomyomas are often asymptomatic during gestation; however, the physiological enlargement of the uterus can outgrow the vascular supply, resulting in hemorrhagic infarction of the leiomyoma (red degeneration). This condition, causing acute abdominal pain, occurs secondary to venous thrombosis, within the periphery of the fibroid [106]. US is the first diagnostic imaging method for suspected complications of fibroids.

The US features of leiomyoma are a spherical, well-defined, solid mass, usually hypoechoic compared to the surrounding myometrium. In red degeneration, US shows heterogeneous or hyperechoic lesions and, later, anechoic components, reflecting cystic changes. The Doppler US demonstrates a decreased or absent flow [107]. MRI could be a useful diagnostic adjunct; multiplanar views allow for the detection of fibroids located deep in the pelvis, enabling differential diagnosis with adnexal masses [20].

Leiomyoma with red degeneration (RDL) may show an unusual signal intensity pattern on MRI. On T1-weighted images, RDL may exhibit diffuse high-signal intensity (Figure 10) or a characteristic peripheral high-intensity rim, due to methemoglobin of blood products confined to thrombosed vessels.

On T2-weighted images, the signal intensity may be variable with a peripheral low-intensity rim, which is also characteristic of RDL; this finding is secondary to the T2*-shortening effects of deoxyhemoglobin, intracellular methemoglobin or hemosiderin of blood products, as well as the expression of venous thrombosis [3,106,107,108]. Takeuchi et al. observed that susceptibility-weighted MR sequences (SWS), which show exquisite sensitivity to blood products, may be useful for the diagnosis of RDL by depicting a characteristic peripheral low-intensity rim [109].

RDL shows a lack of enhancement following gadolinium administration, although contrast is generally not administered during pregnancy [3].

## 5. Vascular Diseases

### Ovarian Vein Thrombosis

Post-partum ovarian vein thrombosis (OVT) is an uncommon but potentially serious disorder during pregnancy and, most often, during puerperium. OVT is reported in approximately 0.05% to 0.18% of vaginal births and in 2% of cases after cesarean delivery [110]. Moreover, this condition occurs in 1–2% of patients with endometritis [7].

Classically, OVT arises in the first 7 days post-partum and is characterized by lower abdominal or flank pain, fever, leucocytosis and, much less commonly, a palpable pelvic mass. Both venous stasis and hypercoagulability place pregnant patients at increased risk of venous thrombosis [111].

OVT is right-sided in almost 90% of cases due to the compression of the inferior vena cava and right ovarian vein by the dextrotated uterus during pregnancy. Other contributory factors include the length of the right ovarian vein, the lack of retrograde flow and multiple incompetent valves. Both ovarian veins may be affected in 14% of cases [7,111].

A prompt diagnosis and treatment of OVT are crucial to avoid potentially life-threatening complications such as septic thrombophlebitis, ovarian infarction, ureteral obstruction, hydronephrosis, renal failure and extension of the thrombus into the inferior vena cava, leading to a pulmonary embolism [112].

US may be employed as a first-line imaging modality, due to its safety, low cost and wide availability.

US features of OVT include an anechoic or hypoechoic tubular structure in the region of the right adnexa and inferior vena cava, with an intra-luminal echogenic mass [113]. Color Doppler shows a decreased or absent flow within the lumen of the vein, depending on whether the thrombosis is partial or complete. An increased flow surrounding the vein may be observed due to the perivascular inflammatory reaction [113].

However, US is an operator-dependent imaging modality and can often be inconclusive due to body habitus or overlying bowel gas, which reduce the detection rate of US [7].

CT with intravenous contrast represents the modality of choice for the diagnosis of OVT post partum, assessing the extent of the thrombosis within the renal vein and the inferior vena cava. A thick-walled, enlarged ovarian vein with rim enhancement and an intra-luminal central filling defect are the main imaging findings of OVT [114]. Multiplanar imaging allows for the distinguishing of the thrombosed vein from the other tubular structures, such as ureters or a loop of bowel (Figure 11).

In a pregnant patient, the ionizing radiation limits the employment of CT [113,114]. Non-contrast MRI can be a problem-solving tool in the clinical suspicion of OVT in pregnancy. On T1-weighted images, the ovarian thrombus can exhibit variable signal intensity depending on the age of luminal blood products, whereas on T2-weighted images, it is shown as a lack of the normal low-signal-intensity flow-void of a patent vessel [81,114]. Nevertheless, images from these unenhanced MRI sequences should be carefully interpreted, because unenhanced sequences are limited by flow signal artifacts [114]. Unenhanced MR venography performed using the time of flight (TOF) technique can depict the thrombus as a flow void in the vessel or the vessel may be completely absent, with or without a surrounding hyperintense signal intensity [3].

## 6. Conclusions

Determining the cause of acute abdominopelvic pain in pregnancy and puerperium is challenging on clinical grounds alone. Urgent imaging is often required, playing an essential role in the assessment of these complications. US followed by MRI are techniques of choice because of their safety profile for both the mother and the foetus. However, US may be limited by altered body habitus, the small field of view and the presence of interfering overlying structures. MRI is extremely accurate in identifying both obstetric and non-obstetric diseases, owing to its superior field of view and soft tissue contrast. CT should be considered in pregnancy on a case-by-case basis if it is likely to offer a high diagnostic yield in problem-solving. Conversely, CT is the imaging modality of choice in patients with post-partum acute abdominal pain, especially after a non-diagnostic US. Comprehensive knowledge of common and uncommon pregnancy and puerperium complications and their imaging features is crucial for accurate diagnosis and early treatment.

## Figures and Tables

**Figure 1 diagnostics-13-02909-f001:**
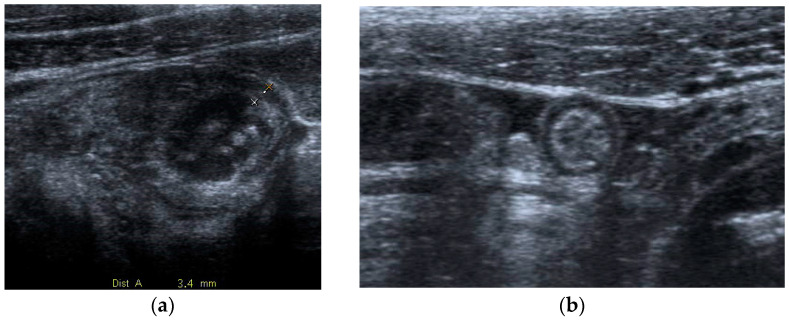
Acute appendicitis in a 29-year-old woman with fever and acute abdominal pain. US shows thick-walled (maximum thickness 3.4 mm as indicated in (**a**)), and dilated tubular structure, containing hyperechoic spots (appendicoliths) (**b**). Periappendiceal fluid is also depicted.

**Figure 2 diagnostics-13-02909-f002:**
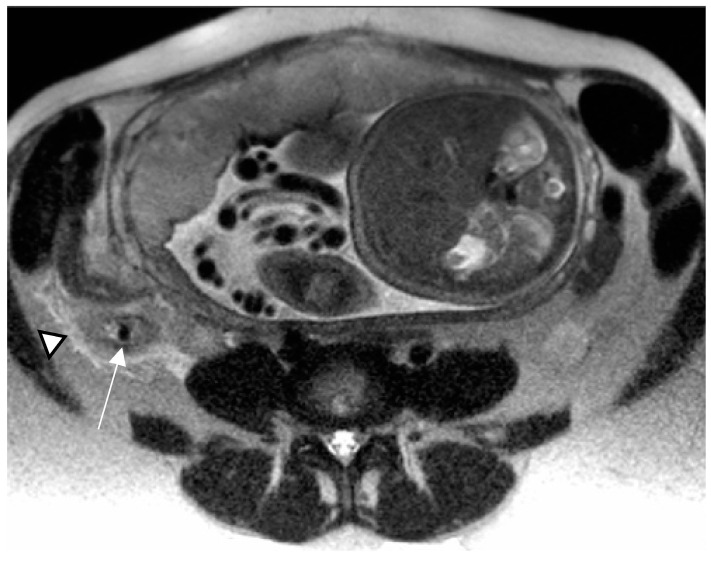
Acute appendicitis in a 32-year-old woman at 34 weeks gestation presenting with right-sided abdominal pain. Axial T2-weighted HASTE image demonstrates a low-signal-intensity appendicolith within a dilated and wall-thickened appendix (arrow). Periappendiceal fat stranding, caused by inflammatory changes, is also depicted (arrowhead).

**Figure 3 diagnostics-13-02909-f003:**
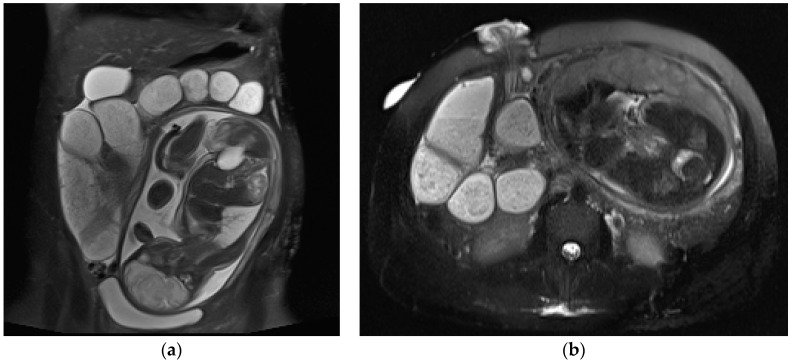
A 21-year-old, third-trimester pregnant woman with known inflammatory bowel disease, presenting with severe abdominal pain. Coronal (**a**) and axial HASTE (**b**) images show multiple dilated fluid-filled bowel loops upstream of the ileostomy in the right iliac fossa. At this level, laparoscopy confirmed the presence of adhesions.

**Figure 4 diagnostics-13-02909-f004:**
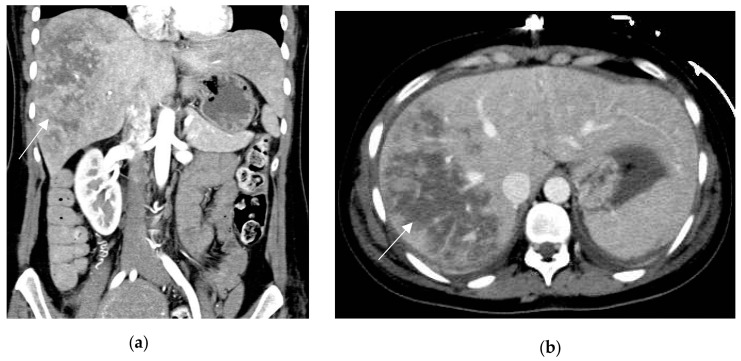
HELLP syndrome in a 29-year-old patient with severe abdominal pain, onset 72 h after delivery. Prepartum hypertension, proteinuria and derangement of LFTs. Coronal and axial contrast-enhanced CT (**a**,**b**) show large and markedly hypoattenuating intrahepatic areas consistent with liver infarction (arrows). Lack of active arterial hemorrhage or intrabdominal bleeding.

**Figure 5 diagnostics-13-02909-f005:**
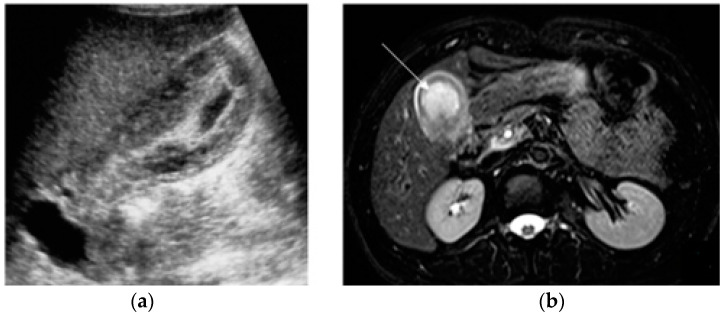
A 24-year-old woman presenting with right upper quadrant pain at 14 weeks of gestation. US (**a**) shows a distended gall bladder with odematous wall and sludge, associated with pericholecystic fat inflammatory changes. On MRI (**b**), the axial T2-weighted fat sat image reveals a thick-walled hydropic gallbladder (arrow), containing sludge and debris. Pericholecystic fat stranding is also depicted.

**Figure 6 diagnostics-13-02909-f006:**
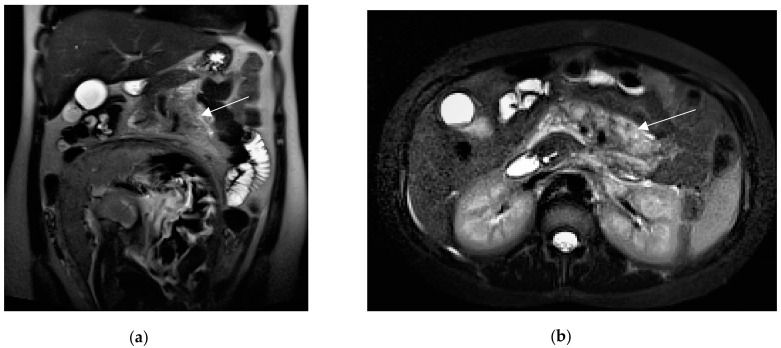
A 27-year-old pregnant woman with acute abdominal pain and elevated lipase levels (451 UI/mL). Coronal (**a**) and axial (**b**) T2-weighted sequences show peripancreatic fat stranding and fluid (arrows), interdigitating through pancreatic parenchyma. These findings are consistent with acute pancreatitis.

**Figure 7 diagnostics-13-02909-f007:**
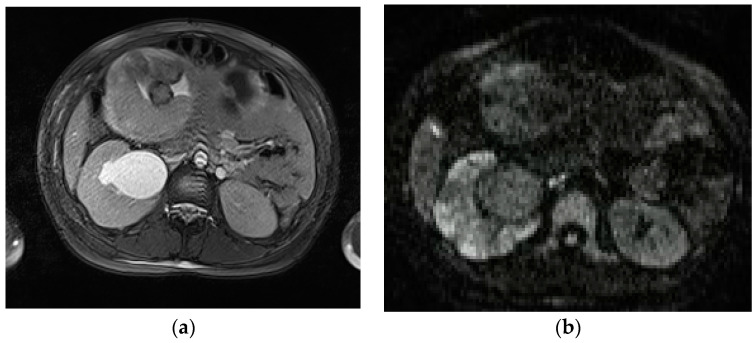
Acute pyelonefritis in a 28-year-old woman at 23 weeks of gestation presenting with right-sided abdominal pain and fever. Axial T2WI fat-sat image (**a**) shows an enlarged right kidney with a dilated pelvis. Axial DWI b-800 (**b**) clearly depicts wedge-shaped areas of high signal intensity in the right kidney, consistent with the foci of nephritis.

**Figure 8 diagnostics-13-02909-f008:**
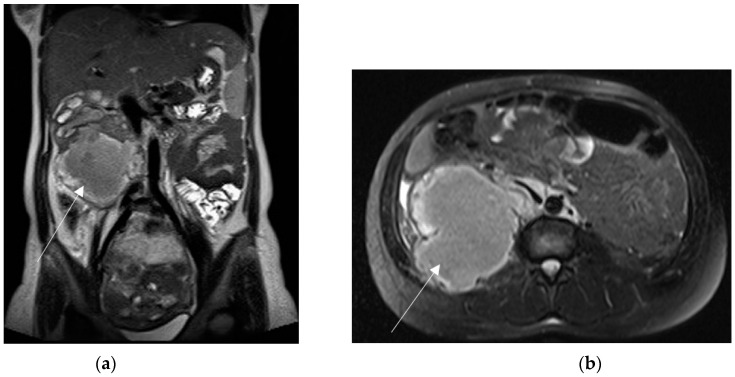
A 21-year-old woman at 20 weeks of gestation was admitted to the hospital with a diagnosis of pyelonephritis and a temperature of 39.4 °C. Coronal T2-weighted HASTE (**a**) and axial T2-weighted fat-sat images (**b**) show a large right-sided and thick-walled fluid collection (arrows), consistent with renal abscess, displacing the kidney.

**Figure 9 diagnostics-13-02909-f009:**
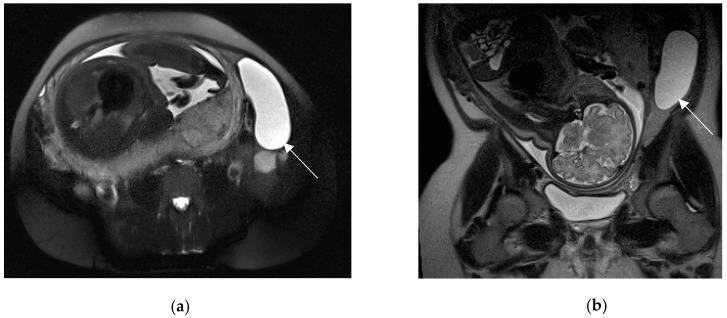
Left adnexal torsion in 28-year-old patient at 35 weeks of gestation. Axial (**a**) and coronal (**b**) T2-weighted images show left cystic formation of about 8 cm (arrow) that has undergone torsion, compressed between the gravid uterus and the abdominal wall.

**Figure 10 diagnostics-13-02909-f010:**
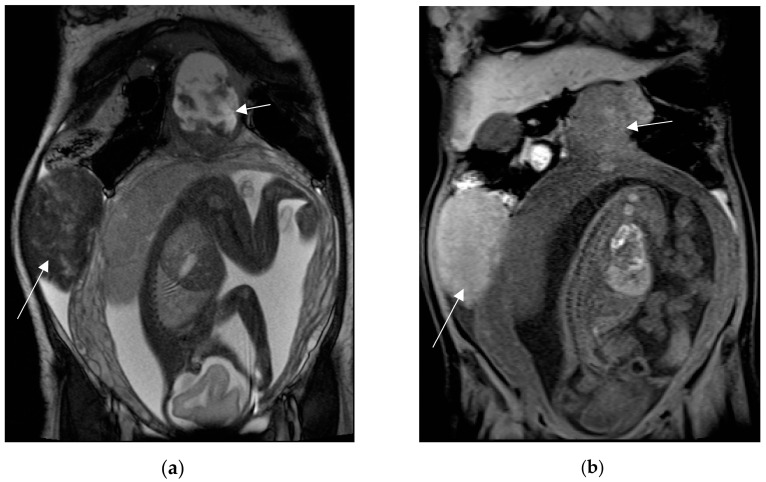
A 44-year-old woman at 27 weeks of gestation, known to have uterine fibroids, presented to the emergency department for acute abdominal pain. MRI shows two large and well-circumscribed pedunculated leiomyomas (arrows), characterized by a variable signal intensity on the coronal T2-weighted image (**a**) and diffuse high-signal intensity on the coronal T1-weighted fat-sat image (**b**). Findings are suggestive of haemorrhage (red degeneration). Peritoneal effusion is also depicted.

**Figure 11 diagnostics-13-02909-f011:**
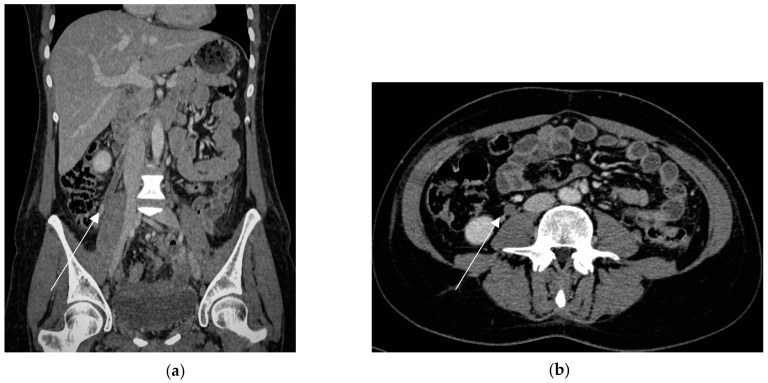
Right ovarian vein thrombosis in a puerperal patient (6 days after delivery) in a 32-year-old woman who underwent hysterectomy for massive primary post-partum hemorrhage due to undiagnosed placenta percreta. Coronal portal venous phase (**a**) shows a low-attenuation tubular structure in the right ovarian vein (arrow) that extends up toward the inferior vena cava, representing thrombosis. Axial portal venous phase (**b**) depicts thrombus as a filling defect in the right ovarian vein (arrow).

## Data Availability

Data sharing is not applicable.

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
