# Peer review of "Imaging of Acute Abdominopelvic Pain in Pregnancy and Puerperium—Part II: Non-Obstetric Complications"

_diagnostics, 2023, doi:10.3390/diagnostics13182909_

Round 1

Reviewer 1 Report

Topic : Imaging diagnosis of acute

Abstract

Line 16 : Ultrasonography

Line 17 : Computed tomography

Line 17-18 : Magnetic resonance imaging 

Keywords: Ultrasonography; Non-obstetric complications

Introduction

Line 43 : Ultrasonography

Line 45 : magnetic resonance imaging 

Line 47 : computed tomography

Line 51-55 : This part should mention what obstetric pain is and what non-obstetric is. And also a short example may be given to link to the second subtopic. 

Contents

Line 59 : 2.1 Acute appendicitis

Line60-61 : occurring in about  0.07% (1 in 1500 pregnancies)

Line 61 : It is very low percentage, why is it the most common?

Line 69 : What is "The ACR Appropriatness Criteria rates US" ?

Line 74 : The sensitivity of US was observed

Line 109 : What is "T2WI"?

Line 127 : 2.2 Small bowel obstruction (SBO)

Line 132: for inflammatory bowel disease (IBD)

Line 133 : The US is the first mode

Line 160 : 2.3 Inflammatory bowel disease

Line 165 : Why underline the word "in pregnancy"?

Line 176 : 3.1 Pregnancy related liver diseases

Line 188 : 3.2 Intrahepatic cholestasis of pregnancy

Line 195,197 : micromoles/L --> µM

Line 201 : 3.3 Acute fatty liver of pregnancy (Some subtopic has abbreviation but some some subtopic no, should write with homogenous pattern)

Line 214 : What is HELLP?

Line 234 : What is " heterogeneously hypo"?

Line 262 : 3.5 Acute cholelithiasis

Line 314 : 3.6 Acute pancreatitis

Line 353 : 4. UROGENITAL TRACT DISEASES

Line 355 : 4.1 Urolithiasis and pyelonephritis

Line 440 : 4.2 Ovarian torsion and adnexal masses

Line 441 : Ovarian torsion (OT)

Line 442 : remove "ovarian torsion"

Line 450 : The US combined with color doppler imaging

Line 498 : 4.3 Uterine leiomyoma

Line 531 : 5. VASCULAR DISEASES

Line 532 : 5.1 Ovarian vein thrombosis

Line 540 : is right sided

Line 554 : What is "ad IVC"

Line 574 : Should add subtopic 6. Conclusion

Minor editing of English language required.

Author Response

Responses to reviewer’s comments

We would like to thank the reviewer for the comments and the careful analysis of our paper. We tried to answer as exhaustively as possible to the comments and modified the text accordingly. We have incorporated most of the suggestions made by the reviewer. Those changes are highlighted within the manuscript. We hope that the final version will meet the required standards.

Reviewer 1

Topic : Imaging diagnosis of acute

We appreciate the proposed change of title, but since these are two Chapters of the same topic (Part 1 Obstetric complication, Part 2 Non obstetric complications), in order to change the title of the second section, the title of the Part1 should be also changed, on which the reviewers presented no objections. We therefore kindly ask the Reviewer to keep the original title. 

Abstract

Line 16 : Ultrasonography

Response: The term “Ultrasound” has been replaced by “Ultrasonography”, throughout the text as suggested.

Line 17 : Computed tomography

Response: The term has been corrected.

Line 17-18 : Magnetic resonance imaging 

Response: The term has been corrected.

Keywords: Ultrasonography; Non-obstetric complications

Response: These keywords have been corrected, as proposed.

Introduction

Line 43 : Ultrasonography

Response: The term “Ultrasound” has been replaced by “Ultrasonography”, throughout the text as suggested.

Line 45 : magnetic resonance imaging

Response: The term has been corrected.  

Line 47 : computed tomography

Response: The term has been corrected.

Line 51-55 : This part should mention what obstetric pain is and what non-obstetric is. And also a short example may be given to link to the second subtopic.

Response: We thank the reviewer for this comment, which improved our introduction. We mentioned the main obstetric (non fetal) and non obstetric causes of acute abdominopelvic pain discussed.

Contents

Line 59 : 2.1 Acute appendicitis

Response: The subtopic title has been corrected accordingly.

Line60-61 : occurring in about  0.07% (1 in 1500 pregnancies). Line 61 : It is very low percentage, why is it the most common?

Response: We are grateful to the reviewer for this comment which gave us the opportunity to provide more information. We added this data related to the prevalence of acute appendicitis in pregnancy. Acute appendicitis is the most common cause of acute abdominal pain in adults and children, so represents also the leading non obstetric surgical emergency in pregnancy. The other non obstetric causes of acute abdomen in pregnancy, such as acute cholecystitis and acute pancreatitis are even rarer.

  1. Kave, F. Parooie, e M. Salarzaei, «Pregnancy and appendicitis: a systematic review and meta-analysis on the clinical use of MRI in diagnosis of appendicitis in pregnant women», World J Emerg Surg, vol. 14, fasc. 1, p. 37, dic. 2019, doi: 10.1186/s13017-019-0254-1.

Line 69 : What is "The ACR Appropriatness Criteria rates US" ?

Response: We thank the reviewer for this comment. The expression “The ACR Appropriatness Criteria rates US" is misleading. Therefore we modified the sentence to clarify that according to the ACR Appropriatness Criteria, ultrasonography is the first line imaging test in the suspicion of appendicitis in pregnancy.

Line 74 : The sensitivity of US was observed

Response: We thank the reviewer for the comment. The sentence has been modified to clarify the meaning.

Line 109 : What is "T2WI"?

Response: We apologize. It is an abbreviation for T2 weighted images. We specified it.

Line 127 : 2.2 Small bowel obstruction (SBO)

Response: The subtopic title has been corrected accordingly.

Line 132: for inflammatory bowel disease (IBD)

Response: The subtopic title has been corrected accordingly. 

Line 133 : The US is the first mode

Response: We corrected the sentence, as suggested.

Line 160 : 2.3 Inflammatory bowel disease

Response: The subtopic title has been corrected, accordingly.  

Line 165 : Why underline the word "in pregnancy"?

Response: We apologize. It was typo, which we corrected.

Line 176 : 3.1 Pregnancy related liver diseases

Response: The section title has been corrected.

Line 188 : 3.2 Intrahepatic cholestasis of pregnancy

Response: The section title has been corrected, accordingly. We have numbered “Intrahepatic cholestasis of pregnancy”, “Acute fatty liver of pregnancy” and HELLP (H = Haemolysis, EL = Elevated Liver enzymes, LP = Low Platelets) syndrome as 3.1.1, 3.1.2, 3.1.3 respectively, as conditions included in “Pregnancy related diseases” (3.1).

Line 195,197 : micromoles/L --> µM

Response: We corrected the abbreviation of unit of measurement as suggested.

Line 201 : 3.3 Acute fatty liver of pregnancy (Some subtopic has abbreviation but some some subtopic no, should write with homogenous pattern)

Response: We thank the reviewer for this comment. We changed the subtopics, using an homogeneous pattern, as suggested.

Line 214 : What is HELLP?

Response: HELLP is an acronym standing for hemolysis, elevated liver enzymes, and low platelets: these three laboratory parameters characterize a variant of severe preeclampsia, representing a life-threatening pregnancy complication. We modified the title of this subtopic to clarify the meaning. 

Line 234 : What is " heterogeneously hypo"?

Response: We are grateful to the reviewer for this comment. As suggested, we have corrected the abbreviation “hypo”, which was misleading, by replacing it with “hypoechoic”. In the manuscript “heterogeneously hypoechoic” refers to US appearance of liver hematoma.

Line 262 : 3.5 Acute cholelithiasis

Response: The subtopic title has been changed, accordingly.

Line 314 : 3.6 Acute pancreatitis

Response: The subtopic title has been corrected, accordingly.

Line 353 : 4. UROGENITAL TRACT DISEASES

Response: The section title has been corrected, as required..

Line 355 : 4.1 Urolithiasis and pyelonephritis

Response: The subtopic title has been corrected, accordingly.

Line 440 : 4.2 Ovarian torsion and adnexal masses

Response: The subtopic title has been corrected, as suggested.

Line 441 : Ovarian torsion (OT)

Response: We inserted the abbreviation (OT) in the first sentence of the subtopic, as suggested.

Line 442 : remove "ovarian torsion"

Response: Thank you for pointing this out. We removed “ovarian torsion” in line 442.

Line 450 : The US combined with color doppler imaging

Response: These words have been corrected, as suggested.

Line 498 : 4.3 Uterine leiomyoma

Response: The subtopic title has been corrected, as suggested.

Line 531 : 5. VASCULAR DISEASES

Response: The section title has been corrected, as suggested.

Line 532 : 5.1 Ovarian vein thrombosis

Response: The subtopic title has been corrected, accordingly.

Line 540 : is right sided

Response: Thank you for pointing this out. The correction has been made.

Line 554 : What is "ad IVC"

Response: We apologize. We removed the abbreviation, which stands for “inferior vena cava”, and we wrote it in full.

Line 574 : Should add subtopic 6. Conclusion

Response: We are grateful to the reviewer for this comment, which gave us the opportunity to improve the quality of our paper. The Conclusion has been added and numbered as 6, accordingly.

Comments on the Quality of English Language

Minor editing of English language required.

Response: We are grateful to the reviewer for this comment. We have edited the English in the manuscript. Those changes are highlighted within the text.

Reviewer 2 Report

Masselli et al present a review of the diagnosis of non-obstetric causes of abdominopelvic pain in pregnancy and postpartum. The manuscript is very well structured and the iconography is rich. I think it can be very useful to the reader. I have some suggestions that could improve the manuscript:

- the addition of a "materials and methods" section, to specify how the bibliographic search was carried out

- a summary table of the main causes of abdominopelvic pain in pregnancy, suggested imaging and expected findings

- Study limitations, if any, and authors' conclusions

Author Response

Responses to reviewer’s comments

Reviewer 2

Masselli et al present a review of the diagnosis of non-obstetric causes of abdominopelvic pain in pregnancy and postpartum. The manuscript is very well structured and the iconography is rich. I think it can be very useful to the reader.

Response: We would like to thank the reviewer for the comments and the critical analysis of our paper. We tried to answer as exhaustively as possible to the comments and modified the text accordingly. We hope that the final version will meet the required standards.

I have some suggestions that could improve the manuscript:

- the addition of a "materials and methods" section, to specify how the bibliographic search was carried out

Response: Thank you for pointing this out. Although we agree that this is an important consideration, the “material and methods” section cannot be added in this manuscript because it is not a systematic review or even a meta-analysis. For the same reason, the “material and methods” section is not included in the first instalment of our review article. We carried out  research on the recent and relevant literature (including the last 15 years) from the PubMed, Cochrane, EMBASE and Scopus databases. The search terms were as follows: Acute abdominopelvic pain; Pregnancy; Post partum; Ultrasonography; Computed Tomography; Magnetic Resonance Imaging; Non-obstetric complications.

- a summary table of the main causes of abdominopelvic pain in pregnancy, suggested imaging and expected findings.

Response: Thank you for this suggestion which gave us the opportunity to improve our paper. We have added a table (Table 1), reporting the main imaging features of the causes of acute abdominopelvic pain in pregnancy discussed in the manuscript.

- Study limitations, if any, and authors' conclusions

Response: We are grateful to the reviewer for this comment. We have added the Conclusion.

Round 2

Reviewer 1 Report

Correction of  all comments.